# CompGS: Efficient 3D Scene Representation via Compressed Gaussian Splatting

## ABSTRACT

Gaussian splatting, renowned for its exceptional rendering quality and efficiency, has emerged as a prominent technique in 3D scene representation. However, the substantial data volume of Gaussian splatting impedes its practical utility in real-world applications. Herein, we propose an efficient 3D scene representation, named Compressed Gaussian Splatting (CompGS), which harnesses compact Gaussian primitives for faithful 3D scene modeling with a remarkably reduced data size. To ensure the compactness of Gaussian primitives, we devise a hybrid primitive structure that captures predictive relationships between each other. Then, we exploit a small set of anchor primitives for prediction, allowing the majority of primitives to be encapsulated into highly compact residual forms. Moreover, we develop a rate-constrained optimization scheme to eliminate redundancies within such hybrid primitives, steering our CompGS towards an optimal trade-off between bitrate consumption and representation efficacy. Experimental results show that the proposed CompGS significantly outperforms existing methods, achieving superior compactness in 3D scene representation without compromising model accuracy and rendering quality. Our code will be released on GitHub for further research.

## CCS CONCEPTS

• **Computing methodologies** → *3D imaging*; *Point-based models*; *Image compression*.

## KEYWORDS

Gaussian splatting, Hybrid primitive structure, Rate-constrained optimization, Compression, 3D scene representation.

## 1 INTRODUCTION

Gaussian splatting (3DGS) [17] has been proposed as an efficient technique for 3D scene representation. In contrast to the preceding implicit neural radiance fields [3, 28, 32], 3DGS [17] intricately depicts scenes by explicit primitives termed 3D Gaussians, and achieves fast rendering through a parallel splatting pipeline [44], thereby significantly prompting 3D reconstruction [7, 13, 26] and view synthesis [21, 39, 40]. Nevertheless, 3DGS [17] requires a considerable quantity of 3D Gaussians to ensure high-quality rendering, typically escalating to millions in realistic scenarios. Consequently,

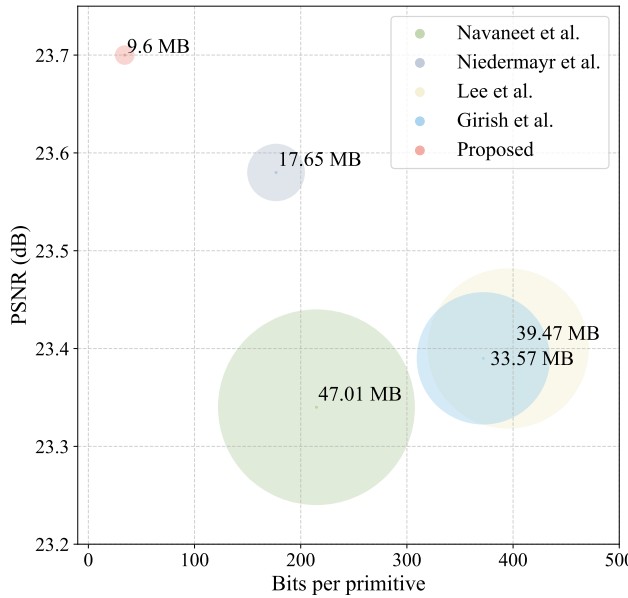

**Figure 1: Comparison between the proposed method and concurrent Gaussian splatting compression methods [10, 20, 33, 34] on the Tanks&Templates dataset [19]. Comparison metrics include rendering quality in terms of PSNR, model size and bits per primitive.**

the substantial burden on storage and bandwidth hinders the practical applications of 3DGS [17], and necessitates the development of compression methodologies.

Recent works [9, 10, 20, 33, 34] have demonstrated preliminary progress in compressing 3DGS [17] by diminishing both quantity and volume of 3D Gaussians. Generally, these methods incorporate heuristic pruning strategies to remove 3D Gaussians with insignificant contributions to rendering quality. Additionally, vector quantization is commonly applied to the retained 3D Gaussians for further size reduction, discretizing continuous attributes of 3D Gaussians into a finite set of codewords. However, extant methods fail to exploit the intrinsic characteristics within 3D Gaussians, leading to inferior compression efficacy, as shown in Figure 1. Specifically, these methods independently compress each 3D Gaussian and neglect the striking local similarities of 3D Gaussians evident in Figure 2, thereby inevitably leaving significant redundancies among these 3D Gaussians. Moreover, the optimization process in these methods solely centers on rendering distortion, which overlooks redundancies within attributes of each 3D Gaussian. Such drawbacks inherently hamper the compactness of 3D scene representations.

This paper proposes Compressed Gaussian Splatting (CompGS), a novel approach that leverages compact primitives for efficient 3D

scene representation. Inspired by the correlations among 3D Gaussians depicted in Figure 2, we devise a hybrid primitive structure that establishes predictive relationships among primitives, to facilitate compact Gaussian representations for scenes. This structure employs a sparse set of anchor primitives with ample reference information for prediction. The remaining primitives, termed coupled primitives, are adeptly predicted by the anchor primitives, and merely contain succinct residual embeddings. Hence, this structure ensures that the majority of primitives are efficiently presented in residual forms, resulting in highly compact 3D scene representation. Furthermore, we devise a rate-constrained optimization scheme to improve the compactness of primitives within the proposed CompGS. Specifically, we establish a primitive rate model via entropy estimation, followed by the formulation of a rate-distortion loss to comprehensively characterize both rendering quality contributions and bitrate costs of primitives. By minimizing this loss, our primitives undergo end-to-end optimization for an optimal rate-distortion trade-off, ultimately yielding advanced compact representations of primitives. Owing to the proposed hybrid primitive structure and the rate-constrained optimization scheme, our CompGS achieves not only high-quality rendering but also compact representations compared to prior works [10, 20, 33, 34], as shown in Figure 1. In summary, our contributions can be listed as follows:

- We propose Compressed Gaussian Splatting (CompGS) for efficient 3D scene representation, which leverages compact primitives to proficiently characterize 3D scenes and achieves an impressive compression ratio up to 110× on prevalent datasets.
- We cultivate a hybrid primitive structure to facilitate compactness, wherein the majority of primitives are adeptly predicted by a limited number of anchor primitives, thus allowing compact residual representations.
- We devise a rate-constrained optimization scheme to further prompt the compactness of primitives via joint minimization of rendering distortion and bitrate costs, fostering an optimal trade-off between bitrate consumption and representation efficiency.

## 2 RELATED WORK

### 2.1 Gaussian Splatting Scene Representation

Kerbl et al. [17] recently proposed a promising technique for 3D scene representation, namely 3DGS. This method leverages explicit primitives to model 3D scenes and renders scenes by projecting these primitives onto target views. Specifically, 3DGS characterizes primitives by 3D Gaussians initialized from a sparse point cloud and then optimizes these 3D Gaussians to accurately represent a 3D scene. Each 3D Gaussian encompasses geometry attributes, i.e., location and covariance, to determine its spatial location and shape. Moreover, appearance attributes, including opacity and color, are involved in the 3D Gaussian to attain pixel intensities when projected to a specific view. Subsequently, the differentiable and highly parallel volume splatting pipeline [44] is incorporated to render view images by mapping 3D Gaussians to the specific view, followed by the optimization of 3D Gaussians via rendering distortion minimization. Meanwhile, an adaptive control strategy is devised

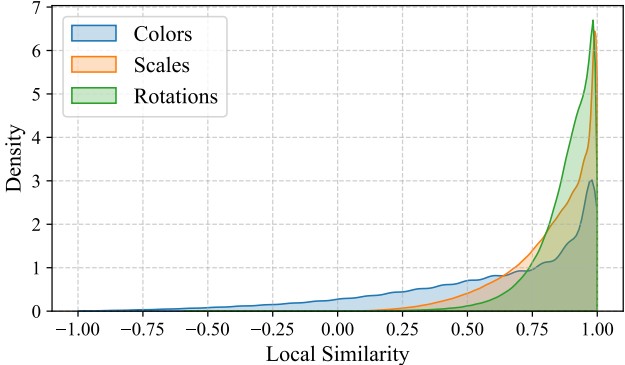

Figure 2: Illustration of local similarities of 3D Gaussians. The local similarity is measured by the average cosine distances between a 3D Gaussian and its 20 neighbors with minimal Euclidean distance.

to adjust the amount of 3D Gaussians, wherein insignificant 3D Gaussians are pruned while crucial ones are densified.

Several methods have been proposed thereafter to improve the rendering quality of 3DGS [17]. Specifically, Yu et al. [41] proposed to apply smoothing filtering to address the aliasing issue in splatting rendering. Hamdi et al. [11] improved 3D Gaussians by generalized exponential functions to facilitate the capability of high-frequency signal fitting. Cheng et al. [8] introduced GaussianPro to improve 3D scene modeling, in which a progressive propagation strategy is designed to effectively align 3D Gaussians with the surface structures of scenes. Huang et al. [15] devised to enhance rendering quality by compensating for projection approximation errors in splatting rendering. Lu et al. [27] developed a structured Gaussian splatting method named Scaffold-GS, in which anchor points are utilized to establish a hierarchical representation of 3D scenes.

However, the rendering benefits provided by Gaussian splatting techniques necessitate maintaining substantial 3D Gaussians, resulting in significant model sizes.

### 2.2 Compressed Gaussian Splatting

Several concurrent works [9, 10, 20, 33, 34] have preliminarily sought to compress models of 3DGS [17], relying on heuristic pruning strategies to reduce the number of 3D Gaussians and quantization to discretize attributes of 3D Gaussians into compact codewords. Specifically, Navaneet et al. [33] designed a Gaussian splatting compression framework named Compact3D. In this framework, K-means-based vector quantization is leveraged to quantize attributes of 3D Gaussians to discrete codewords, thereby reducing the model size of 3DGS [17]. Niedermayr et al. [34] proposed to involve sensitivities of 3D Gaussians during quantization to alleviate quantization distortion, and leveraged entropy coding to reduce statistical redundancies within codewords. Lee et al. [20] devised learnable masks to reduce the quantity of 3D Gaussians by eliminating non-essential ones, and introduced grid-based neural fields to compactly model appearance attributes of 3D Gaussians. Furthermore, Fan et al. [9] devised a Gaussian splatting compression framework named LightGaussian, wherein various technologies are

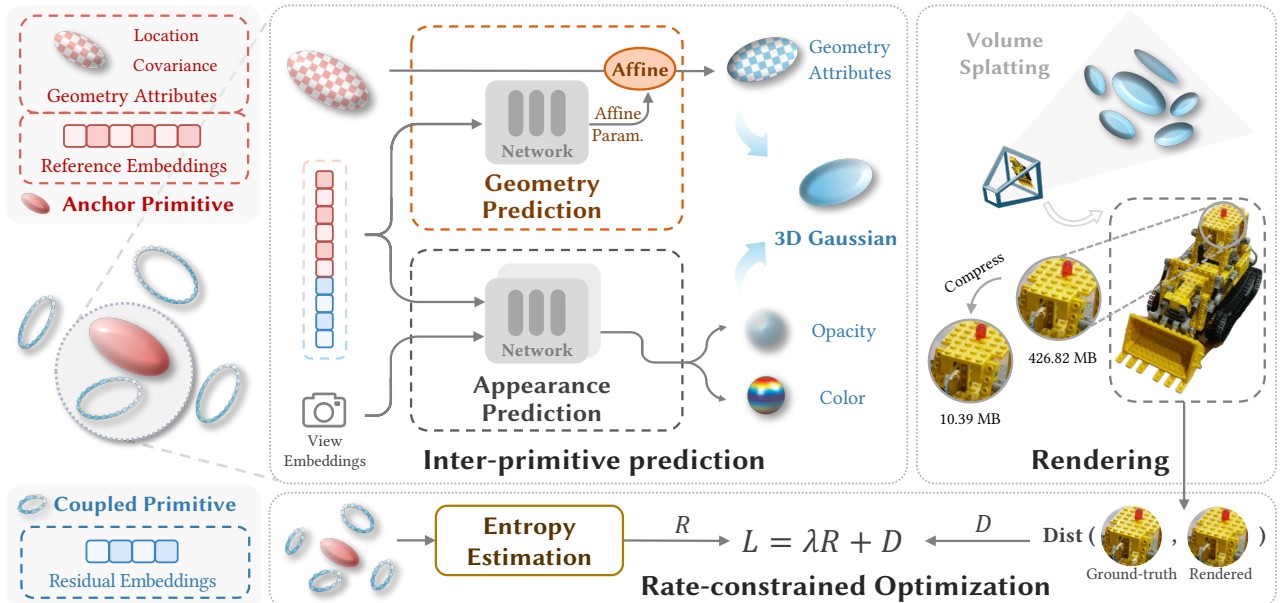

**Figure 3: Overview of our proposed method.**

combined to reduce model redundancies within 3DGS [17]. Notably, a distillation paradigm is designed to effectively diminish the size of color attributes within 3D Gaussians. Girish et al. [10] proposed to represent 3D Gaussians by compact latent embeddings and decode 3D Gaussian attributes from the embeddings.

However, these existing methods optimize 3D Gaussians merely by minimizing rendering distortion, and then independently compress each 3D Gaussian, thus leaving substantial redundancies within obtained 3D scene representations.

## 2.3 Video Coding

Video coding, an outstanding data compression research field, has witnessed remarkable advancements over the past decades and cultivated numerous invaluable coding technologies. The most advanced traditional video coding standard, versatile video coding (VVC) [6], employs a hybrid coding framework, capitalizing on predictive coding and rate-distortion optimization to effectively reduce redundancies within video sequences. Specifically, predictive coding is devised to harness correlations among pixels to perform prediction. Subsequently, only the residues between the original and predicted values are encoded, thereby reducing pixel redundancies and enhancing compression efficacy. Notably, VVC [6] employs affine transform [25] to improve prediction via modeling non-rigid motion between pixels. Furthermore, VVC [6] employs rate-distortion optimization to adaptively configure coding tools, hence achieving superior coding efficiency.

Recently, neural video coding has emerged as a competitive alternative to traditional video coding. These methods adhere to the hybrid coding paradigm, integrating neural networks for both prediction and subsequent residual coding. Meanwhile, end-to-end optimization is employed to optimize neural networks within compression frameworks via rate-distortion cost minimization. Within the neural video coding pipeline, entropy models, as a vital component of residual coding, are continuously improved to accurately estimate the probabilities of residues and, thus, the rates. Specifically, Ballé et al. [1] proposed a factorized entropy bottleneck that utilizes fully-connected layers to model the probability density function of the latent codes to be encoded. Subsequently, Ballé et al. [2] developed a conditional Gaussian entropy model, with hyper-priors extracted from latent codes, to parametrically model the probability distributions of the latent codes. Further improvements concentrate on augmenting prior information, including spatial context models [24, 29, 43], channel-wise context models [16, 30], and temporal context models [14, 22, 23].

In this paper, motivated by the advancements of video coding, we propose to employ the philosophy of both prediction and rate-distortion optimization to effectively eliminate redundancies within our primitives.

## 3 METHODOLOGY

### 3.1 Overview

As depicted in Figure 3, the proposed CompGS encompasses a hybrid primitive structure for compact 3D scene representation, involving anchor primitives to predict attributes of the remaining coupled primitives. Specifically, a limited number of anchor primitives are created as references. Each anchor primitive $\omega$ is embodied by geometry attributes (location $\mu_{\omega}$ and covariance $\Sigma_{\omega}$) and reference embeddings $f_{\omega}$. Then, $\omega$ is associated with a set of $K$ coupled primitives $\{\gamma_1, \ldots, \gamma_K\}$, and each coupled primitive $\gamma_k$ only includes compact residual embeddings $g_k$ to compensate for prediction errors. In the following inter-primitive prediction, the geometry attributes of $\gamma_k$ are obtained by warping the corresponding anchor primitive $\omega$ via affine transform, wherein affine parameters are adeptly predicted by $f_{\omega}$ and $g_k$. Concurrently, the

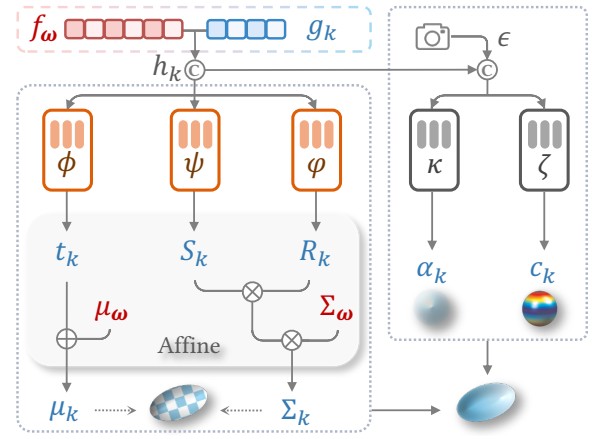

**Figure 4: Illustration of the proposed inter-primitive prediction.**

view-dependent appearance attributes of $\gamma_k$, i.e., color and opacity, are predicted using $\{f_\omega, g_k\}$ and view embeddings [27]. Owing to the hybrid primitive structure, the proposed CompGS can compactly model 3D scenes by redundancy-eliminated primitives, with the majority of primitives presented in residual forms.

Once attaining geometry and appearance attributes, these coupled primitives can be utilized as 3D Gaussians to render view images via volume splatting [28]. In the subsequent rate-constrained optimization, rendering distortion $D$ can be derived by calculating the quality degradation between the rendered and corresponding ground-truth images. Additionally, entropy estimation is exploited to model the bitrate of anchor primitives and associated coupled primitives. The derived bitrate $R$, along with the distortion $D$, are used to formulate the rate-distortion cost $\mathcal{L}$. Then, all primitives within the proposed CompGS are jointly optimized via rate-distortion cost minimization, which facilitates the primitive compactness and, thus, compression efficiency. The optimization process of our primitives can be formulated by

$$\Omega^*, \Gamma^* = \arg\max_{\Omega,\Gamma} \mathcal{L} = \arg\max_{\Omega,\Gamma} \lambda R + D, \quad (1)$$

where $\lambda$ denotes the Lagrange multiplier to control the trade-off between rate and distortion, and $\{\Omega, \Gamma\}$ denote the set of anchor primitives and coupled primitives, respectively.

### 3.2 Inter-primitive Prediction

The inter-primitive prediction is proposed to derive the geometry and appearance attributes of coupled primitives based on associated anchor primitives. As a result, coupled primitives only necessitate succinct residues, contributing to compact 3D scene representation. As shown in Figure 4, the proposed inter-primitive prediction takes an anchor primitive $\omega$ and an associated coupled primitive $\gamma_k$ as inputs, and predicts geometry and appearance attributes for $\gamma_k$, including location $\mu_k$, covariance $\Sigma_k$, opacity $\alpha_k$, and color $c_k$. Specifically, residual embeddings $g_k$ of $\gamma_k$ and reference embeddings $f_\omega$ of $\omega$ are first fused by channel-wise concatenation, yielding prediction features $h_k$. Subsequently, the geometry attributes $\{\mu_k, \Sigma_k\}$ are generated by warping $\omega$ using affine transform [25], with the

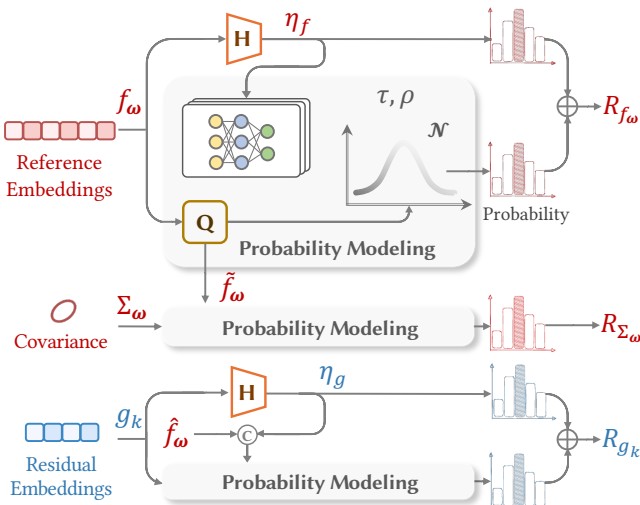

**Figure 5: Illustration of the proposed entropy estimation.**

affine parameters $\beta_k$ derived from $h_k$ via learnable linear layers. This process can be formulated as

$$\mu_k, \Sigma_k = \mathcal{A}(\mu_\omega, \Sigma_\omega | \beta_k), \quad (2)$$

where $\mathcal{A}$ denotes the affine transform, and $\{\mu_\omega, \Sigma_\omega\}$ denote location and covariance of the anchor primitive $\omega$, respectively. To improve the accuracy of geometry prediction, $\beta_k$ is further decomposed into translation vector $t_k$, scaling matrix $S_k$, and rotation matrix $R_k$, which are predicted by neural networks, respectively, i.e.,

$$t_k = \phi(h_k), \quad S_k = \psi(h_k), \quad R_k = \varphi(h_k), \quad (3)$$

where $\{\phi(\cdot), \psi(\cdot), \varphi(\cdot)\}$ denote the neural networks. Correspondingly, the affine process in Equation 2 can be further formulated as

$$\mu_k = \mu_\omega + t_k, \quad \Sigma_k = S_k R_k \Sigma_\omega. \quad (4)$$

Simultaneously, to model the view-dependent appearance attributes $\alpha_k$ and $c_k$, view embeddings $\epsilon$ are generated from camera poses and concatenated with prediction features $h_k$. Then, neural networks are employed to predict $\alpha_k$ and $c_k$ based on the concatenated features. This process can be formulated by

$$\alpha_k = \kappa(\epsilon \oplus h_k), \quad c_k = \zeta(\epsilon \oplus h_k), \quad (5)$$

where $\oplus$ denotes the channel-wise concatenation and $\{\kappa(\cdot), \zeta(\cdot)\}$ denote the neural networks for color and opacity prediction, respectively.

### 3.3 Rate-constrained Optimization

The rate-constrained optimization scheme is devised to achieve compact primitive representation via joint minimization of bitrate consumption and rendering distortion. As shown in Figure 5, we establish the entropy estimation to effectively model the bitrate of both anchor and coupled primitives. Specifically, scalar quantization [1] is first applied to $\{\Sigma_\omega, f_\omega\}$ of anchor primitive $\omega$ and $g_k$ of

**Table 1: Performance comparison on the Tanks&Templates dataset [19].**

| Methods | PSNR (dB) | SSIM | LPIPS | Size (MB) |
|---|---|---|---|---|
| Kerbl et al. [17] | 23.72 | 0.85 | 0.18 | 434.38 |
| Navaneet et al. [33] | 23.34 | 0.84 | 0.19 | 47.01 |
| Niedermayr et al. [34] | 23.58 | 0.85 | 0.19 | 17.65 |
| Lee et al. [20] | 23.40 | 0.84 | 0.20 | 39.47 |
| Girish et al. [10] | 23.39 | 0.84 | 0.20 | 33.57 |
| Proposed | 23.70 | 0.84 | 0.21 | 9.60 |
| | 23.39 | 0.83 | 0.22 | 7.27 |
| | 23.11 | 0.81 | 0.24 | 5.89 |

associated coupled primitive $\gamma_k$, i.e.,

$$\hat{\Sigma}_\omega = Q(\frac{\Sigma_\omega}{s_\Sigma}), \quad \hat{f}_\omega = Q(\frac{f_\omega}{s_f}), \quad \hat{g}_k = Q(\frac{g_k}{s_g}), \tag{6}$$

where $Q(\cdot)$ denotes the scalar quantization and $\{s_\Sigma, s_f, s_g\}$ denote the corresponding quantization steps. However, the rounding operator within $Q$ is not differentiable and breaks the back-propagation chain of optimization. Hence, quantization noises [1] are utilized to simulate the rounding operator, yielding differentiable approximations as

$$\tilde{\Sigma}_\omega = \delta_\Sigma + \frac{\Sigma_\omega}{s_\Sigma}, \quad \tilde{f}_\omega = \delta_f + \frac{f_\omega}{s_f}, \quad \tilde{g}_k = \delta_f + \frac{g_k}{s_g}, \tag{7}$$

where $\{\delta_\Sigma, \delta_f, \delta_g\}$ denote the quantization noises obeying uniform distributions. Subsequently, the probability distribution of $\tilde{f}_\omega$ is estimated to calculate the corresponding bitrate. In this process, the probability distribution $p(\tilde{f}_\omega)$ is parametrically formulated as a Gaussian distribution $\mathcal{N}(\tau_f, \rho_f)$, where the parameters $\{\tau_f, \rho_f\}$ are predicted based on hyperpriors [2] extracted from $f_\omega$, i.e.,

$$p(\tilde{f}_\omega) = \mathcal{N}(\tau_f, \rho_f), \quad \text{with} \ \ \tau_f, \rho_f = \mathcal{E}_f(\eta_f), \tag{8}$$

where $\mathcal{E}_f$ denotes the parameter prediction network and $\eta_f$ denotes the hyperpriors. Moreover, the probability of hyperpriors $\eta_f$ is estimated by the factorized entropy bottleneck [1], and the bitrate of $f_\omega$ can be calculated by

$$R_{f_\omega} = \mathbb{E}_\omega \left[ -\log p(\tilde{f}_\omega) - \log p(\eta_f) \right], \tag{9}$$

where $p(\eta_f)$ denotes the estimated probability of hyperpriors $\eta_f$. Furthermore, $\tilde{f}_\omega$ is used as contexts to model the probability distributions of $\tilde{\Sigma}_\omega$ and $\tilde{g}_k$. Specifically, the probability distribution of $\tilde{\Sigma}_\omega$ is modeled by Gaussian distribution with parameters $\{\tau_\Sigma, \rho_\Sigma\}$ predicted by $\tilde{f}_\omega$, i.e.,

$$p(\tilde{\Sigma}_\omega) = \mathcal{N}(\tau_\Sigma, \rho_\Sigma), \quad \text{with} \ \ \tau_\Sigma, \rho_\Sigma = \mathcal{E}_\Sigma(\tilde{f}_\omega), \tag{10}$$

where $p(\tilde{\Sigma}_\omega)$ denotes the estimated probability distribution and $\mathcal{E}_\Sigma$ denotes the parameter prediction network for covariance. Meanwhile, considering the correlations between the $f_\omega$ and $g_k$, the probability distribution of $\tilde{g}_k$ is modeled via Gaussian distribution conditioned on $\tilde{f}_\omega$ and extracted hyperpriors $\eta_g$, i.e.,

$$p(\tilde{g}_k) = \mathcal{N}(\tau_g, \rho_g), \quad \text{with} \ \ \tau_g, \rho_g = \mathcal{E}_g(\tilde{f}_\omega \oplus \eta_g), \tag{11}$$

**Table 2: Performance comparison on the Deep Blending dataset [12].**

| Methods | PSNR (dB) | SSIM | LPIPS | Size (MB) |
|---|---|---|---|---|
| Kerbl et al. [17] | 29.54 | 0.91 | 0.24 | 665.99 |
| Navaneet et al. [33] | 29.89 | 0.91 | 0.25 | 72.46 |
| Niedermayr et al. [34] | 29.45 | 0.91 | 0.25 | 23.87 |
| Lee et al. [20] | 29.82 | 0.91 | 0.25 | 43.14 |
| Girish et al. [10] | 29.90 | 0.91 | 0.25 | 61.69 |
| Proposed | 29.69 | 0.90 | 0.28 | 8.77 |
| | 29.40 | 0.90 | 0.29 | 6.82 |
| | 29.30 | 0.90 | 0.29 | 6.03 |

**Table 3: Performance comparison on the Mip-NeRF 360 dataset [4].**

| Methods | PSNR (dB) | SSIM | LPIPS | Size (MB) |
|---|---|---|---|---|
| Kerbl et al. [17] | 27.46 | 0.82 | 0.22 | 788.98 |
| Navaneet et al. [33] | 27.04 | 0.81 | 0.23 | 86.10 |
| Niedermayr et al. [34] | 27.12 | 0.80 | 0.23 | 28.61 |
| Lee et al. [20] | 27.05 | 0.80 | 0.24 | 49.60 |
| Girish et al. [10] | 27.04 | 0.80 | 0.24 | 65.09 |
| Proposed | 27.26 | 0.80 | 0.24 | 16.50 |
| | 26.78 | 0.79 | 0.26 | 11.02 |
| | 26.37 | 0.78 | 0.28 | 8.83 |

where $p(\tilde{g}_k)$ denotes the estimated probability distribution. Accordingly, the bitrate of $\Sigma_\omega$ and $g_k$ can be calculated by

$$\begin{aligned} R_{\Sigma_\omega} &= \mathbb{E}_\omega \left[ -\log p(\tilde{\Sigma}_\omega) \right], \\ R_{g_k} &= \mathbb{E}_{\gamma_k} \left[ -\log p(\tilde{g}_k) - \log p(\eta_g) \right], \end{aligned} \tag{12}$$

where $p(\eta_g)$ denotes the probability of $\eta_g$ estimated via the factorized entropy bottleneck [1]. Consequently, the bitrate consumption of the anchor primitive $\omega$ and its associated $K$ coupled primitives $\{\gamma_1, \ldots, \gamma_K\}$ can be further calculated by

$$R_{\omega,\gamma} = R_{f_\omega} + R_{\Sigma_\omega} + \sum_{k=1}^{K} R_{g_k}. \tag{13}$$

Furthermore, to formulate the rate-distortion cost depicted in Equation 1, the rate item $R$ is calculated by summing bitrate costs of all anchor and coupled primitives, and the distortion item $D$ is provided by the rendering loss [17]. Then, the rate-distortion cost is used to perform end-to-end optimization of primitives and neural networks within the proposed method, thereby attaining high-quality rendering under compact representations.

### 3.4 Implementation Details

In the proposed method, the dimension of reference embeddings is set to 32, and that of residual embeddings is set to 8. Neural networks used in both prediction and entropy estimation are implemented by two residual multi-layer perceptrons. Quantization steps $\{s_f, s_g\}$ are fixed to 1, whereas $s_\Sigma$ is a learnable parameter

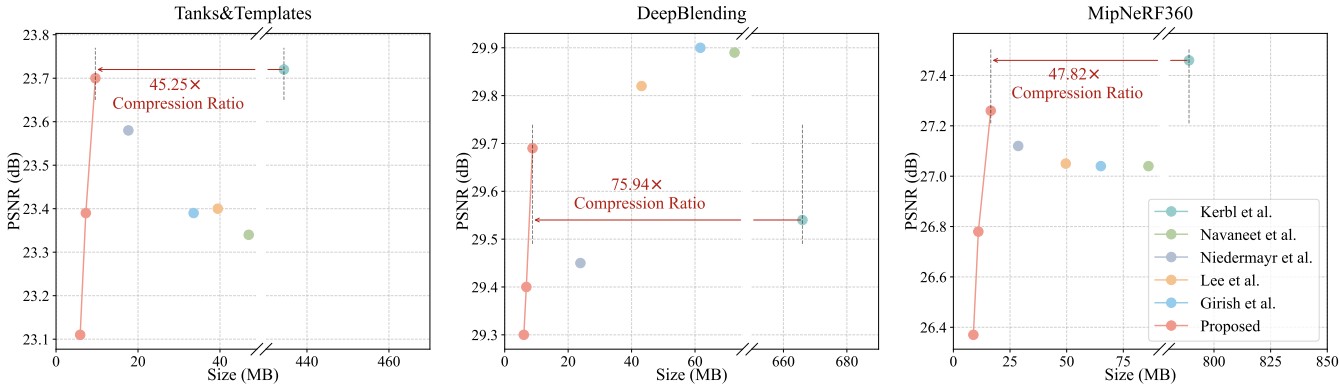

**Figure 6: Rate-distortion curves of the proposed method and comparison methods [10, 17, 20, 33, 34].**

with an initial value of 0.01. The Lagrange multiplier $\lambda$ in Equation 1 is set to $\{0.001, 0.005, 0.01\}$ to obtain multiple bitrate points. Moreover, the anchor primitives are initialized from sparse point clouds produced by voxel-downsampled SfM points [36], and each anchor primitive is associated with $K = 10$ coupled primitives. After the optimization, reference embeddings and covariance of anchor primitives, along with residual embeddings of coupled primitives, are compressed into bitstreams by arithmetic coding [31], wherein the probability distributions are provided by the entropy estimation module. Additionally, point cloud codec G-PCC [37] is employed to compress locations of anchor primitives.

The proposed method is implemented based on PyTorch [35] and CompressAI [5] libraries. Adam optimizer [18] is used to optimize parameters of the proposed method, with a cosine annealing strategy for learning rate decay. Additionally, adaptive control [27] is applied to manage the number of anchor primitives, and the volume splatting [44] is implemented by custom CUDA kernels [17].

## 4 EXPERIMENTS

### 4.1 Experimental Settings

**Datasets.** To comprehensively evaluate the effectiveness of the proposed method, we conduct experiments on three prevailing view synthesis datasets, including Tanks&Templates [19], Deep Blending [12] and Mip-NeRF 360 [4]. These datasets comprise high-resolution multiview images collected from real-world scenes, characterized by unbounded environments and intricate objects. Furthermore, we conform to the experimental protocols in 3DGS [17] to ensure evaluation fairness. Specifically, the scenes specified by 3DGS [17] are involved in evaluations, and the sparse point clouds provided by 3DGS [17] are utilized to initialize our anchor primitives. Additionally, one view is selected from every eight views for testing, with the remaining views used for training.

**Comparison methods.** We employ 3DGS [17] as an anchor method and compare several concurrent compression methods [10, 20, 33, 34]. To retrain these models for fair comparison, we adhere to their default configurations as prescribed in corresponding papers. Notably, extant compression methods [10, 20, 33, 34] only provide the configuration for a single bitrate point. Moreover, each method

undergoes five independent evaluations in a consistent environment to mitigate the effect of randomness, and the average results of the five experiments are reported. Additionally, the detailed results with respect to each scene are provided in the Appendix.

**Evaluation metrics.** We adopt PSNR, SSIM [38] and LPIPS [42] to evaluate the rendering quality, alongside model size, for assessing compression efficiency. Meanwhile, we use training, encoding, decoding, and view-average rendering time to quantitatively compare the computational complexity across various methods.

### 4.2 Experimental Results

**Qualitative Results.** The proposed method achieves the highest compression efficiency on the Tanks&Templates dataset [19], as illustrated in Table 1. Specifically, compared with 3DGS [17], our method achieves a significant compression ratio, ranging from 45.25× to 73.75×, with a size reduction up to 428.49 MB. These results highlight the effectiveness of our proposed CompGS. Moreover, our method surpasses existing compression methods [10, 20, 33, 34], with the highest rendering quality, i.e., 23.70 dB, and the smallest bitstream size. This advancement stems from comprehensive utilization of the hybrid primitive structure and the rate-constrained optimization, which effectively facilitate compact representations of 3D scenes.

Table 2 shows the quantitative results on the Deep Blending dataset [12]. Compared with 3DGS [17], the proposed method achieves remarkable compression ratios, from 75.94× to 110.45×. Meanwhile, the proposed method realizes a 0.15 dB improvement in rendering quality at the highest bitrate point, potentially attributed to the integration of feature embeddings and neural networks. Furthermore, the proposed method achieves further bitrate savings compared to existing compression methods [10, 20, 33, 34]. Consistent results are observed on the Mip-NeRF 360 dataset [4], wherein our method considerably reduces the bitrate consumption, down from 788.98 MB to at most 16.50 MB, correspondingly, culminating in a compression ratio up to 89.35×. Additionally, our method demonstrates a remarkable improvement in bitrate consumption over existing methods [10, 20, 33, 34]. Notably, within the *Stump* scene of the Mip-NeRF 360 dataset [4], our method significantly reduces the model size from 1149.30 MB to 6.56 MB, achieving **an**

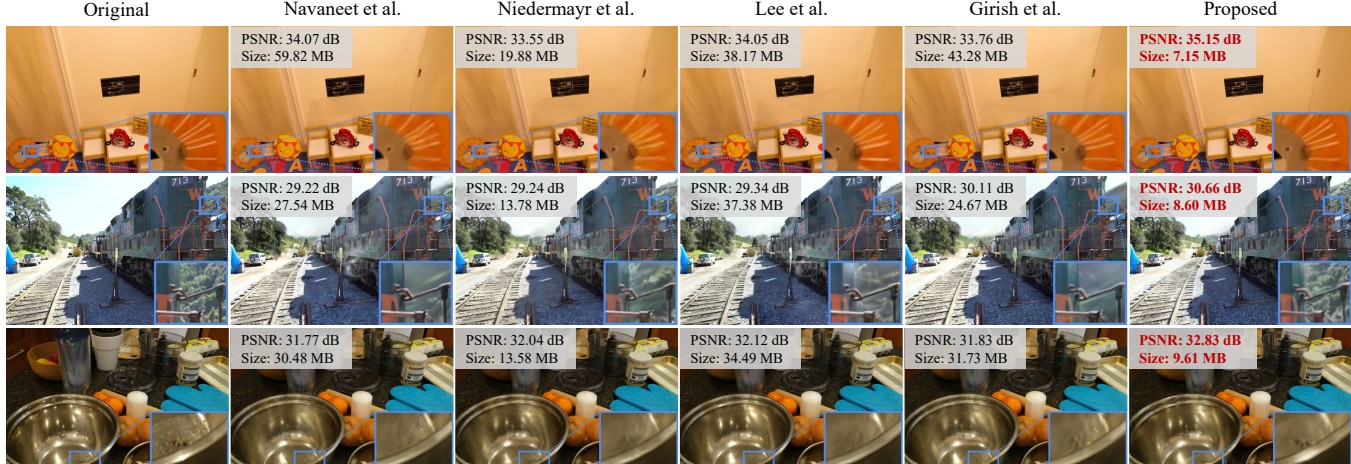

Figure 7: Qualitative results of the proposed method compared to existing compression methods [10, 20, 33, 34].

Table 4: Ablation studies on the Tanks&Templates dataset [19].

| Hybrid Primitive Structure | Rate-constrained Optimization | Train | | | | Truck | | | |
|---|---|---|---|---|---|---|---|---|---|
| | | PSNR (dB) | SSIM | LPIPS | Size (MB) | PSNR (dB) | SSIM | LPIPS | Size (MB) |
| ✗ | ✗ | 22.02 | 0.81 | 0.21 | 257.44 | 25.41 | 0.88 | 0.15 | 611.31 |
| ✓ | ✗ | 22.15 | 0.81 | 0.23 | 48.58 | 25.20 | 0.86 | 0.19 | 30.38 |
| ✓ | ✓ | 22.12 | 0.80 | 0.23 | 8.60 | 25.28 | 0.87 | 0.18 | 10.61 |

**extraordinary compression ratio of 175.20×**. This exceptional outcome demonstrates the effectiveness of the proposed method and its potential for practical implementation of Gaussian splatting schemes. Moreover, we present the rate-distortion curves to intuitively demonstrate the superiority of the proposed method. It can be observed from Figure 6 that our method achieves remarkable size reduction and competitive rendering quality as compared to other methods [10, 17, 20, 33, 34]. Detailed performance comparisons for each scene are provided in the Appendix to further substantiate the advancements realized by the proposed method.

**Qualitative Results**. Figure 7 illustrates the qualitative comparison of the proposed method and other compression methods [10, 20, 33, 34], with specific details zoomed in. It can be observed that the rendered images obtained by the proposed method exhibit clearer textures and edges.

## 4.3 Ablation Studies

**Effectiveness on hybrid primitive structure.** The hybrid primitive structure is proposed to exploit a limited number of anchor primitives to proficiently predict attributes of the remaining coupled primitives, thus enabling an efficient representation of these coupled primitives by compact residual embeddings. To verify the effectiveness of the hybrid primitive structure, we incorporate it into the baseline 3DGS [17], and the corresponding results on the Tanks&Templates dataset [19] are depicted in Table 4. It can be observed that the hybrid primitive structure greatly prompts the

compactness of 3D scene representations, exemplified by a reduction of bitstream size from 257.44 MB to 48.58 MB for the *Train* scene and from 611.31 MB down to 30.38 MB for the *Truck* scene. This is because the devised hybrid primitive structure can effectively eliminate the redundancies among primitives, thus achieving compact 3D scene representation.

Furthermore, we provide bitstream analysis of our method on the *Train* scene in Figure 8. It can be observed that the bit consumption of coupled primitives is close to that of anchor primitives across multiple bitrate points, despite the significantly higher number of coupled primitives compared to anchor primitives. Notably, the average bit consumption of coupled primitives is demonstrably lower than that of anchor primitives, which benefits from the compact residual representation employed by the coupled primitives. These findings further underscore the superiority of the hybrid primitive structure in achieving compact 3D scene representation.

**Effectiveness on rate-constrained optimization.** The rate-constrained optimization is devised to effectively improve the compactness of our primitives via minimizing the rate-distortion loss. To evaluate its effectiveness, we incorporate it with the hybrid primitive structure, establishing the framework of our proposed method. As shown in Table 4, the employment of rate-constrained optimization leads to a further reduction of the bitstream size from 48.58 MB to 8.60 MB for the *Train* scene, equal to an additional bitrate reduction of 82.30%. On the *Truck* scene, a substantial decrease of 65.08% in bitrate is achieved. The observed bitrate efficiency can be

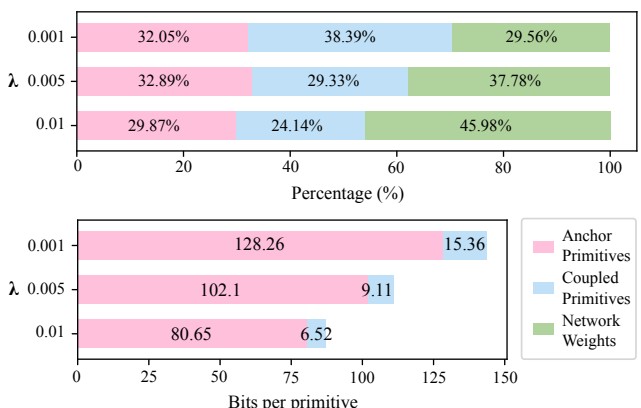

**Figure 8: Bitstream analysis at multiple bitrate points. The upper figure illustrates the proportion of different components within bitstreams, and the bottom figure quantifies the bit consumption per anchor primitive and per coupled primitive.**

**Table 5: Ablation studies on the residual embeddings.**

|  | PSNR (dB) | SSIM | LPIPS | Size (MB) |
|---|---|---|---|---|
| w.o. Res. Embed. | 20.50 | 0.73 | 0.31 | 5.75 |
| Proposed | 21.49 | 0.78 | 0.26 | 5.51 |

attributed to the capacity of the proposed method to learn compact primitive representations through rate-constrained optimization.

**Effectiveness of Residual embeddings.** Recent work [27] introduces a primitive derivation paradigm, whereby anchor primitives are used to generate new primitives. To demonstrate our superiority over this paradigm, we devise a variant, named "w.o. Res. Embed.", which adheres to such primitive derivation paradigm [27] by removing the residual embeddings within coupled primitives. The experimental results on the *Train* scene of Tanks&Templates dataset [19], as shown in Table 5, reveal that, this variant fails to obtain satisfying rendering quality and inferiors to our method. This is because such indiscriminate derivation of coupled primitives can hardly capture unique characteristics of coupled primitives. In contrast, our method can effectively represent such characteristics by compact residual embeddings.

**Proportion of coupled primitives.** We conduct ablations on the *Train* scene from the Tanks&Templates dataset [19] to investigate the impact of the proportion of coupled primitives. Specifically, we adjust the proportion of coupled primitives by manipulating the number of coupled primitives $K$ associated with each anchor primitive. As shown in Table 6, the case with $K = 10$ yields the best rendering quality, which prompts us to set $K$ to 10 in our experiments. Besides, the increase of $K$ from 10 to 15 leads to a rendering quality degradation of 0.22 dB. This might be because excessive coupled primitives could lead to an inaccurate prediction.

**Table 6: Ablation studies on the proportion of coupled primitives.**

| K | PSNR (dB) | SSIM | LPIPS | Size (MB) |
|---|---|---|---|---|
| 5 | 22.04 | 0.80 | 0.24 | 7.87 |
| 10 | 22.12 | 0.80 | 0.23 | 8.60 |
| 15 | 21.90 | 0.80 | 0.24 | 8.28 |

**Table 7: Complexity comparison on the Tanks&Templates dataset [19].**

| Methods | Train (min) | Enc-time (s) | Dec-time (s) | Render (ms) |
|---|---|---|---|---|
| Navaneet et al. [33] | 14.38 | 68.29 | 12.32 | 9.88 |
| Niedermayr et al. [34] | 15.50 | 2.23 | 0.25 | 9.74 |
| Lee et al. [20] | 44.70 | 1.96 | 0.18 | 6.60 |
| Girish et al. [10] | 8.95 | 0.54 | 0.64 | 6.96 |
| Proposed | 37.83 | 6.27 | 4.46 | 5.32 |

## 4.4 Complexity Analysis

Table 7 reports the complexity comparisons between the proposed method and existing compression methods [10, 20, 33, 34] on the Tanks&Templates dataset [19]. In terms of training time, the proposed method requires an average of 37.83 minutes for training, which is shorter than the method proposed by Lee et al. [20] and longer than other methods. This might be attributed to that the proposed method needs to optimize both primitives and neural networks. Additionally, the encoding and decoding times of the proposed method are both less than 10 seconds, which illustrates the practicality of the proposed method for real-world applications. In line with comparison methods, the per-view rendering time of the proposed method averages 5.32 milliseconds, due to the utilization of highly-parallel splatting rendering algorithm [44].

## 5 CONCLUSION

This work proposes a novel 3D scene representation method, Compressed Gaussian Splatting (CompGS), which utilizes compact primitives for efficient 3D scene representation with remarkably reduced size. Herein, we tailor a hybrid primitive structure for compact scene modeling, wherein coupled primitives are proficiently predicted by a limited set of anchor primitives and thus, encapsulated into succinct residual embeddings. Meanwhile, we develop a rate-constrained optimization scheme to further improve the compactness of primitives. In this scheme, the primitive rate model is established via entropy estimation, and the rate-distortion cost is then formulated to optimize these primitives for an optimal trade-off between rendering efficacy and bitrate consumption. Incorporated with the hybrid primitive structure and rate-constrained optimization, our CompGS outperforms existing compression methods, achieving superior size reduction without compromising rendering quality.

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
