# OpenReview forum: "CompGS: Efficient 3D Scene Representation via Compressed Gaussian Splatting"
_acmmm.org/ACMMM/2024/Conference — MM2024 Poster_

### Official Review · Reviewer_gJsm · 2024-05-10

**Rating:** 4
**Confidence:** 3

**Summary:**

In this paper, the authors proposed a novel method for 3D Gaussian Splatting compression. Inspired by the field of video coding, they categorized all Gaussian primitives within a scene into anchor primitives and their corresponding coupled primitives, using a residual way for inter-primitive prediction. Moreover, they applied a rate-constrained optimization for bitrate control. The experimental results demonstrate that the proposed method surpassed existing pioneer works for Gaussian splatting compression.

**Strengths:**

In this paper, the idea of predicting coupled primitives from the residual embeddings and the corresponding anchor primitive is of great novelty, but still with practical flaws. Moreover, the rate control method borrowed from image and point cloud compression modeling the quantization as a Gaussian noise is also solid. The experiments are sufficient and effectively demonstrate the advancement of the proposed method.

**Limitations:**

There are still some mistakes in this paper. For example, in Eq. (1), this should be a $\arg\min$ problem rather than an $\arg\max$ problem to minimize the bitrate $R$ and the distortion $D$; and in Fig. 5, the symbols $H$ and $Q$ are not specified. In addition, there are some details of the proposed method are not explained. The authors mentioned that $K$ coupled primitives are determined for each anchor primitive; how are these coupled primitives determined? And how are the embeddings of these Gaussian primitives determined?

**Suitability:**

2

---

### Official Review · Reviewer_PyhK · 2024-05-24

**Rating:** 4
**Confidence:** 3

**Summary:**

This paper presents a novel method to compress 3D Gaussian Splat Scenes. It represents a 3DGS through a set of anchor primitives while each anchor is associated with a set of coupled primitives, where the geometry and appearance attributes are predicted by the neural network. Finally, a rate-constrained optimization scheme further prompts the compactness. The experiment results show that CompGS achieves higher compression ratio without a significant drop in visual quality.

**Strengths:**

1. This paper is well written with good figures and tables.
2. The hybrid structure and rate-constrained optimization scheme significantly reduce the size of a 3DGS scene, from hundreds of MBs to several MBs.
3. The experiments are sufficient. It shows the improvement of hybrid structure, the improvement of hybrid structure + rate-constrained optimization scheme, the impact of the number of coupled primitives, and so on.

**Limitations:**

1. It would be better to assign three names to the three values obtained from "Proposed" in Table 1-3. That would be much clearer for readers to understand the table.
2. Is there any explanation for why the PSNR of some compressed scenes is higher than original 3DGS?
3. How many anchor primitives are selected for each scene? How will the number of anchor primitives affect the compression results?

**Suitability:**

3

---

### Official Review · Reviewer_xQs9 · 2024-05-25

**Rating:** 4
**Confidence:** 1

**Summary:**

This paper proposed CompGS for 3D Gaussian compression, enabling efficient 3D scene representation. The authors introduced a hybrid primitive structure and a rate-constrained optimization scheme to conduct typical prediction and rate distortion optimization for 3D Gaussian point cloud. Experimental results demonstrate that CompGS outperforms some 3D Gaussian compression methods in terms of compression efficiency.

**Strengths:**

The paper's organization is well-structured.

The authors have conducted a thorough evaluation using a diverse set of comparison datasets.

**Limitations:**

The paper proposes the use of typical Point Cloud Compression ideas, such as geometry and attribute compression, for the 3D Gaussian point cloud representation. However, the prediction and rate-distortion optimization approaches employed are rather commonplace in the point cloud compression literature.

The primary concern is the lack of comparisons to existing point cloud compression methods. Without evaluating the performance of the proposed approach against established point cloud compression techniques, it is difficult to assess the true novelty and contributions of this work.

[1] GRASP-Net: Geometric residual analysis and synthesis for point cloud compression[C]//Proceedings of the 1st International Workshop on Advances in Point Cloud Compression, Processing and Analysis. 2022:
[2] Sparse tensor-based multiscale representation for point cloud geometry compression[J]. IEEE Transactions on Pattern Analysis and Machine Intelligence, 2022.

**Suitability:**

3

---

### Official Review · Reviewer_zvdE · 2024-05-26

**Rating:** 4
**Confidence:** 2

**Summary:**

This paper proposed an efficient 3D scene representation, with tailored hybrid primitive structure and rate-constrained optimization scheme, achieving significant compression ratio.

**Strengths:**

- The paper reads well and is substantiated with adequate experiments and ablation studies. The evaluation is detailed with significant analysis/results for PSNR, SSIM and LPIPS and meaningful illustrations.
- The work stemmed from the observation of local similarities of 3D Gaussians and proposed to use anchor primitives and prediction network to represent the 3DGS, which is solid and enlightening.

**Limitations:**

- In Section 3.2 Implementation Details, the rationale of hyperparameter settings should be clarified.
- The number of The proportion of coupled primitives K has important impact on the compact representation, and related analysis was made in the ablation study section. Nevertheless, it would be important to provide "guidance" (e.g., mathematical model) on how to set the value K.
- The sparse point clouds were utilized to initialize the anchor primitives, and G-PCC was used to compress the geometry of the anchor primitives. As a point cloud codec, G-PCC can achieve significant compression ratio. Therefore, to better illustrate the performance of the proposed method, listing the size and compression ratio with and without G-PCC would be appreciated.

**Suitability:**

3

---

### Meta-Review · Area_Chair_nMie · 2024-07-08

**Recommendation:** Accept (Poster)
**Confidence:** 5

**Metareview:**

This paper presents an efficient 3D scene representation, Compressed Gaussian Splatting (CompGS), which harnesses compact Gaussian primitives for faithful 3D scene modeling with a remarkably reduced data size. The experiments are sufficient and effectively demonstrate the performance of the proposed method. All reviewers have positive final ratings, are satisfied with the response, and recommend accepting the paper. I agree with their recommendation. Thanks for the authors' effort and rebuttal.